# Endothelial Effects of Simultaneous Expression of Human HO-1, E5NT, and ENTPD1 in a Mouse

**DOI:** 10.3390/ph16101409

**Published:** 2023-10-04

**Authors:** Paulina Mierzejewska, Noemi Di Marzo, Magdalena A. Zabielska-Kaczorowska, Iga Walczak, Ewa M. Slominska, Marialuisa Lavitrano, Roberto Giovannoni, Barbara Kutryb-Zajac, Ryszard T. Smolenski

**Affiliations:** 1Department of Biochemistry, Medical University of Gdansk, Debinki 1 St., 80-211 Gdansk, Poland; paulina.mierzejewska@gumed.edu.pl (P.M.); magdalena.zabielska@gumed.edu.pl (M.A.Z.-K.); igawalczak@gumed.edu.pl (I.W.); eslom@gumed.edu.pl (E.M.S.); 2School of Medicine and Surgery, University of Milano-Bicocca, 20900 Monza, Italy; noemi.dimarzo91@gmail.com (N.D.M.); marialuisa.lavitrano@unimib.it (M.L.); roberto.giovannoni@unipi.it (R.G.); 3Department of Physiology, Medical University of Gdansk, Debinki 1 St., 80-211 Gdansk, Poland; 4Department of Biology, University of Pisa, 56026 Pisa, Italy

**Keywords:** adenosine, ecto-5′-nucleotidase, ecto-nucleoside triphosphate diphosphohydrolase 1, nucleotide metabolism, xenotransplantation, endothelium

## Abstract

The vascular endothelium is key target for immune and thrombotic responses that has to be controlled in successful xenotransplantation. Several genes were identified that, if induced or overexpressed, help to regulate the inflammatory response and preserve the transplanted organ function and metabolism. However, few studies addressed combined expression of such genes. The aim of this work was to evaluate in vivo the effects of the simultaneous expression of three human genes in a mouse generated using the multi-cistronic F2A technology. Male 3-month-old mice that express human heme oxygenase 1 (hHO-1), ecto-5′-nucleotidase (hE5NT), and ecto-nucleoside triphosphate diphosphohydrolase 1 (hENTPD1) (Transgenic) were compared to wild-type FVB mice (Control). Background analysis include extracellular nucleotide catabolism enzymes profile on the aortic surface, blood nucleotide concentration, and serum L-arginine metabolites. Furthermore, inflammatory stress induced by LPS in transgenic and control mice was used to characterize interleukin 6 (IL-6) and adhesion molecules endothelium permeability responses. Transgenic mice had significantly higher rates of extracellular adenosine triphosphate and adenosine monophosphate hydrolysis on the aortic surface in comparison to control. Increased levels of blood AMP and adenosine were also noticed in transgenics. Moreover, transgenic animals demonstrated the decrease in serum monomethyl-L-arginine level and a higher L-arginine/monomethyl-L-arginine ratio. Importantly, significantly decreased serum IL-6, and adhesion molecule levels were observed in transgenic mice in comparison to control after LPS treatment. Furthermore, reduced endothelial permeability in the LPS-treated transgenic mice was noted as compared to LPS-treated control. The human enzymes (hHO-1, hE5NT, hENTPD1) simultaneously encoded in transgenic mice demonstrated benefits in several biochemical and functional aspects of endothelium. This is consistent in use of this approach in the context of xenotransplantation.

## 1. Introduction

Transplantation is an effective approach to managing patients with organ failure, but it is still a challenge for modern medicine. The increasing incidence of vital organ failure, combined with their insufficient availability, results in an extremely long time for patients on waiting lists [1]. A revolution in the treatment of organ failure seems to be the transplantation of vascularized organs across the species barrier. Xenotransplantation, i.e., replacing human failing organs with animal organs, could effectively solve organ shortages [2]. Despite the many similarities between human and porcine organs regarding size and physiology, the main obstacles to clinical application are among others immune rejection between species [3].

The vascular endothelium, which lies at the interface between the recipient’s blood and the transplanted donor organ, bears the full brunt of the immunological processes [4]. Endothelial cells (ECs) create a single cell layer that lines the inside of all lymphatic and blood vessels and controls the exchange of oxygen and nutrients between blood and tissues/organs. The vascular endothelium is a very active organ involved in the regulation of cellular migration and adhesion, vascular tone, vessel wall permeability, coagulation, and various inflammatory processes [5]. Moreover, endothelium participates in the transport of immune cells across the organism to reach organs and tissues regulating immune surveillance in physiological conditions as well as infectious complications and metastasis [6]. In the physiological state, EC tight junctions regulate the homeostasis of tissues and organs and paracellular diffusion. However, during inflammation, crucial changes occur to the junction ultrastructures, which allows the entry of immune cells [4].

One approach, that seems promising in the context of xenotransplant rejection, is to genetically modify the donor organs to prevent or reduce attack by the human immune system, and thus achieve compatibility with the human organism. Over the years several genes, whose induction or over-expression can regulate the inflammatory response and preserve the transplanted organs’ metabolism were identified [7]. Due to their immunosuppressive, cytoprotective, and anti-inflammatory properties, extracellular adenosine, a metabolite of adenine nucleotides, may be of particular interest in the field of xenotransplantation. Extracellular adenosine is produced in the pathway controlled mainly by ecto-nucleotidases, including ecto-nucleoside triphosphate diphosphohydrolase 1 (ENTPD1 or CD39), and ecto-5′-nucleotidase (E5NT or CD73). ENTPD1 converts the extracellular Adenosine Triphosphate (ATP)/Adenosine Diphosphate (ADP) to Adenosine Monophosphate (AMP). Then, E5NT, which is bound onto the cell membrane via a glycosyl-phosphatidylinositol anchor, with the catalytic site facing the extracellular region, hydrolyzes AMP to adenosine [8]. Many reports, as well as our results, support the concept of the crucial role of the maintenance of the extracellular adenosine optimal concentration in endothelium homeostasis [9,10]. E5NT is found in a variety of tissues, including the kidney, liver, lung, brain, and heart, as well as on leukocytes, thymus, spleen, and lymph nodes. This extensive presence of this ecto-enzyme throughout the organism is linked to its high abundance in endothelium [11]. E5NT activity is crucial for proper endothelium functioning because the E5NT gene deletion leads to the development of age-dependent endothelial dysfunction in mice [9]. ENTPD1 is the other dominant ecto-nucleotidase expressed by endothelial cells [12]. ENTPD1 has a protective role against platelet aggregation and vessel occlusion by terminating the proinflammatory and prothrombotic effects of ATP and ADP [13]. In turn, heme oxygenase 1 (HO-1) was shown to be critical for the endothelium as it is engaged in the conversion of heme to carbon monoxide (CO), free iron, and biliverdin, which have anti-inflammatory, anti-apoptotic, and antioxidant effects [14]. In addition to its known role in protection against oxidative stress, HO-1 inhibits the expression of proinflammatory genes associated with endothelial cell activation [15].

This work aimed to evaluate the protective effects of the simultaneous expression of a novel combination of anti-inflammatory human genes, HO-1, E5NT, and ENTPD1, in in vivo mice model using F2A technology [7,16].

## 2. Results

To evaluate if the simultaneous expression of hHO-1, hE5NT, and hENTPD1 affects the vascular metabolism of adenine nucleotides, the rates of ATP and AMP hydrolysis and adenosine deamination on the aortas of transgenic and control mice were measured (Figure 1).

The ATP and AMP hydrolysis rates on the aortic surface of the transgenic mice were approximately two times higher in comparison with the control (Figure 1a,b), which was an expected effect due to the genetic modifications of these animals. There was no statistically significant change in the vascular adenosine deamination rate between both study groups (Figure 1c).

To assess the cell energy balance changes, the blood nucleotide concentration was measured (Figure 2). There were no significant changes in blood ATP, ADP, but also nicotinamide adenine dinucleotide (NAD) concentration in transgenic mice in comparison to the control (Figure 2a,b,e). However, we observed approximately two-fold increased concentration of AMP and adenosine in transgenic mice as compared to control (Figure 2c,d).

To check, if the application of exogenous hHO-1, hE5NT, and hENTPD1 proteins in mice affects the metabolism, especially in the context of the endothelium, the concentration of plasma L-arginine and its metabolites and derivatives were evaluated. There was a tendency to higher L-arginine concentration and its analog—asymmetric dimethyl L-arginine (ADMA) in transgenic mice as compared to control. Moreover, transgenic mice were characterized by significantly decreased plasma level of other L-arginine analog—L-NG-monomethyl arginine (L-NMMA) in comparison to control. Although there were no differences in plasma ornithine and citrulline between study groups of animals (Table 1), there was also a tendency to lower Ornithine/Citrulline ratio and considerable decrease in Ornithine/Arginine ratio in transgenic mice as compared to controls.

Considering that the activities of the three, used in this model, proteins have been reported to exert anti-inflammatory effects [17,18,19], their potential role in protection against acute LPS-induced inflammation was investigated. The level of both adhesion molecules, associated with the endothelium activation—ICAM-1 and VCAM-1, as well as IL-6, were increased in both study groups of animals, regardless of their genotype as compared to non-treated animals. However, we observed a considerable decrease in IL-6, ICAM-1, and VCAM-1 in LPS-treated transgenic mice in comparison to LPS-treated controls (Figure 3).

Furthermore, vascular endothelial function was assessed by Evans Blue (EB) permeability assay. The EB concentration was significantly increased in serum but also in the aortas of LPS-treated mice, both transgenic and controls, as compared to non-treated animals. Interestingly, LPS-treated transgenic mice were characterized by considerably lower EB concentration in serum and aortic tissue extract in comparison to LPS-treated controls (Figure 4).

## 3. Discussion

This work validated strategy for combined expression of three human genes, HO-1, E5NT, and ENTPD1 in transgenic animals and proved benefits for the function of vascular endothelium. We have shown attenuation of the effect of the LPS-induced inflammation on the parameters associated with endothelial function such as concentration of adhesion molecules or endothelial permeability. Each human gene used in this work has been reported to have anti-inflammatory, anti-apoptotic, or antioxidant properties when overexpressed or induced in the cells or organisms.

The three human genes coding sequences were included in an expression cassette that allowed the simultaneous translation of all proteins starting from a single mRNA by the application of the F2A technology. In earlier work, De Giorgi M et al. verified the activity of the vector. They demonstrated that the three proteins encoded by the multi-cistronic cassette were fully functional and allowed the increased formation of their enzymatic products [7].

However, the most important finding of this work is that the transgenic mice with the simultaneous expression of the human HO-1, E5NT, and ENTPD1 are characterized by the decreased proinflammatory IL-6 and lower levels of adhesion molecules (ICAM-1 and VCAM-1) linked with the activation of ECs, as well as diminished Evans Blue concentration in the serum and aortic tissue extract after induction of the acute inflammation with LPS. One of the possible, protective for the endothelium, mechanisms in this model is the orientation of the extracellular metabolism of adenine nucleotides to the increased production of adenosine. The animals used in this work were characterized by an increased ATP hydrolysis rate, corresponding to higher ENTPD1 activity [20] and AMP hydrolysis, for which E5NT is responsible [21]. Moreover, there were no differences in vascular adenosine deamination rate between study groups, which may additionally contribute to maintaining high concentrations of E5NT-derived adenosine on the vascular surface of these transgenic animals.

Extracellular adenosine is considered an important signaling molecule, which exerts its effects by the activation of the P1 receptors: A1, A2A, A2B, and A3. Each of these receptors acts via various signal transduction mechanisms [22]. Thompson et al. demonstrated that decreased production of extracellular adenosine is associated with a dramatic vascular leakage, most likely resulting from diminished activation of the A2B receptor on the endothelium. They suggested that the adenosine released at sites affected by the inflammation may contribute to the reduction in the swelling that is prominent at inflamed sites [11]. It is well known that E5NT-derived adenosine diminishes leukocyte recruitment, inhibits activated neutrophil adhesion to the endothelium, as well as neutrophil-induced injury to the vascular endothelium [23]. Moreover, adenosine exerts regulatory effects on M1 macrophages, which are mediated by A2A and A2B receptors [24]. It has been shown that adenosine inhibits TNF-alfa, but also IL-6 and IL-12 production and augments anti-inflammatory IL-10 release by LPS-or bacteria-activated macrophages mostly via A2A receptors [25]. However, adenosine reduced LPS-induced TNF-alpha formation even in A2A deficient mice, with stimulation of A2B receptors [24]. Furthermore, adenosine promotes M2 macrophage activation, which is considered an anti-inflammatory factor [26].

Moreover, it has been reported that E5NT deficiency affected endothelial permeability. The exacerbation of hypoxia-induced vascular leak in, e.g., lungs and kidneys in response to the E5NT activity reduction was observed [27]. It also has been shown that in vitro co-culture experiments involving ECs and T lymphocytes demonstrated that genetic deletion of E5NT leads to the transendothelial migration of T lymphocytes and enhanced the expression of TNF-alpha, VCAM-1, and INF-gamma [28]. In addition, C. St Hilaire et al. observed a significant increase in vascular calcification in patients with NT5E gene mutation encoding E5NT [29]. Our earlier studies support these findings. We have previously shown, that the E5NT deletion contributes to the development of age-dependent endothelial dysfunction in mice. This damage was related to disturbances in L-arginine metabolism [9]. We also observed the development of aortic valve dysfunction in E5NT-deficient mice, evidenced by the functional and structural changes consistent with aortic stenosis [21]. Additionally, the existing literature indicates that extracellular adenosine may have a protective role in the context of implant adaptation. Tuskamoto H. et al. demonstrated that the E5NT deficiency leads to heightened severity of graft-versus-host disease (GVHD). Both E5NT-/- mice, as well as mice subjected to a pharmacological inhibition of E5NT exhibited increased cytotoxicity, cell expansion, and higher concentrations of proinflammatory cytokines [30]. Moreover, Lee SC. et al. successfully produced two healthy hE5NT transgenic cloned piglets with normal reproductive ability for the next generation. They assumed that these pigs may be useful for the production of “multi-gene” hybrid transgenic animals, which will be more effective model to use in research for controlling hyperacute and acute vascular rejection after xenotransplantation [31].

Besides induced E5NT activity, stimulation of ENTPD1 activity also leads to the maintenance of the extracellular adenosine on the vascular endothelium. This ecto-nucleotidase has been lately reported as an important molecule in the regulation of maintaining the antithrombotic properties of the endothelium [32]. Through hydrolyzing ATP to ADP and to AMP, ENTPD1 reduces ADP levels at the inflammation site and regulates both P2 and P1 receptor activation. Thus, it has been considered a crucial modulator of thrombus formation. ENTPD1-deficient mice demonstrated a strong prothrombotic phenotype and in the model of ischemia-reperfusion injury, that develop a large infarct size [33,34]. In addition, located on the surface of endothelial cells and circulating platelets, ENTPD1 contributes to the suppressive function of human and mouse regulatory T cells (Tregs) [PMID: 17502665]. Ni X. et al. demonstrated that ENTPD1 signaling plays an important role in mediating the protective impact of gingival-derived MSCs (GMSCs)—a stem-cell population found in human gingival tissue—during acute GVHD. GMSCs that were treated with an ENTPD1 inhibitor (POM-1) prior to transfer demonstrated a reduced therapeutic effect in acute GVHD. Moreover, they showed that GMSCs not only express ENTPD1 but also enhance the frequency of ENTPD1+Foxp3+ Tregs—a unique type of cells with a regulatory function in the autoimmune disease model. That supports the production of adeno-sine and promotes immune suppression of effector T cells in vitro and in vivo [35]. Moreover, mice lacking ENTPD1 activity develop pulmonary arterial hypertension, and remodeling, as well as ventricular hypertrophy, following hypoxia [36]. Knight J. S et al. also suggested that ENTPD1 may have a protective role in vascular disease associated with lupus. In the experimental mice model of lupus, the suppression of neutrophil extracellular trap release mediated by ENTPD1 was observed [37]. All of these findings support the hypothesis, that both of the extracellular ecto-nucleotidases, ENTPD1 and E5NT, represent an important regulation point of the inflammation, contributing to maintaining endothelial permeability and anti-thrombotic properties.

In this work, we focused our studies on the protective role of overexpression of ecto-nucleotidases and extracellular adenosine in the context of endothelial protection. However, our mice were also characterized by the induction of hHO-1 activity. He Y et al. demonstrated a protective role of HO-1 in renal ischemia-reperfusion injury. HO-1 deficient mice were characterized by the upregulation of VCAM-1 together with the augmentation of leukocyte recruitment and inflammatory damage. HO-1 knockout also increases neutrophil adherence and vascular membrane basement migration in vitro [38]. Kang L et al. also indicated the upregulation and vasoprotective effects of HO-1, as well as the vasorelaxant effects of CO in the partial carotid artery ligation model of pathological shear stress [39]. Moreover, in children without a functional HO-1 allele, increased development of atherosclerosis was observed [40]. It was also noticed that both HO-1 products—CO and biliverdin—inhibit neointima formation [41,42]. Considering these data, HO-1 induction may also be responsible for the protective effects of our tricistronic vector, used in transgenic animals. Additionally, hHO-1 has already been successfully used in xenotransplantation. Singh AK et al. described clinical cardiac transplant of “multi-gene” pig heart with hHO-1 and other nine gene edits, along with the use of a novel non-ischemic continuous perfusion preservation system. Genetics of donor pig was GTKO, B4GalNT2KO, CMAHKO, GHRKO, hCD46, hDAF, hTBM, hEPCR, hCD47, and hHO-1 [43]

In conclusion, this study provides the first experimental evidence for the anti-inflammatory effects of the combination of three human genes: HO-1, E5NT, and ENTPD1 simultaneously expressed in a mice model through an F2A system-based multicistronic technique. The application of this tricistronic vector in vivo exerts protective effects on the vascular endothelium, such as diminished levels of adhesion molecules linked with endothelial cell activation and decreased endothelium permeability in inflammatory conditions. This genetic modification may be beneficial to preserve endothelial function in the context of xenotransplantation.

## 4. Materials and Methods

### 4.1. Reagents

Adenosine (A4036), AMP (01930), ATP (A26209), Evans blue (E2129), HBSS (H9269), LPS (E. coli 055:b5, L2880), and phosphate-buffered saline (PBS) (P5493), as well as the soluble ICAM-1, VCAM-1, and IL-6 serum levels (RAB0220, RAB0506, RAB0308) were purchased from Sigma-Aldrich/Merck (Saint Louis, MO, USA).

### 4.2. Triple Cistronic Plasmid Construction and Transgenic Mice Production

Transgenic mice overexpressing hHO-1/hE5NT/hENTPD1 were produced by Takis, genetically modified mice were then housed for a few generations at Allevamenti Plaisant mouse facility and then transferred to Department of Biochemistry, Medical University of Gdansk for colony expansion and experimental procedures. A triple cistronic plasmid containing the coding sequences of exogenous human Heme Oxygenase 1 (hHO-1), Ecto-5′-nucleotidase (hE5NT), and Ecto-nucleoside triphosphate diphosphohydrolase 1 (hENTPD) was assembled by a series of PCR reactions and cloning steps as previously reported [7]. The final triple cistronic vector thus produced was named pCX-TRI-2A (Addgene plasmid #74674; http://n2t.net/addgene:74674, accessed on 2 October 2023; RRID:Addgene_74674). The pCX-TRI-2A construct was tested in stably transfected murine embryonic fibroblast, NIH3T3 cells [7]. Before DNA microinjection, the pCX-TRI-2A construct was digested to excise the tricistronic cassette from the vector backbone and the transgene fragment was microinjected on FVB/N mouse strain zygotes for the production of transgenic animals. For genetic characterization, diagnostic PCRs analyses were performed on genomic DNA extracted from mouse tail biopsies. Each founder mouse has been crossed with wild-type FVB/N mice to produce hemizygous transgenic mice. Diagnostic PCRs were performed on 75 ng of genomic DNA to investigate the presence or absence of exogenous construct integrated into the mice genome. Specific oligonucleotides, targeting transgenic hE5NT (Intern-hCD73Fw, TGTTGGTGATGAAGTTGTGG; hCD73Rev, CGCCAACTTGAGAAGGTCAAAA) and hHO1 (5′hHO1Fw, CTGGAGGAGGAGATTGAGCG; hHO1Rev, CGCCAACTTGAGAAGGTCAAAA) sequences, were designed either manually or using Primer Express 3.0 Software [44] to allow discrimination between transgenic and wild-type mice. PCR products were run under electric field on 1.2% agarose gel for amplicon size analysis. Only animals positive for both PCR reactions were considered transgenic and employed for subsequent crosses.

### 4.3. Animal Maintenance and Treatment

FVB Wild Type (Control) and FVB hHO-1/hE5NT/hENTPD1 overexpressing (Transgenic) mice, both aged three months, were used in this study. Water and a standard chow diet were provided ad libitum. All experiments were conducted following the Guide for the Care and Use of the Laboratory Animals published by the European Parliament, Directive 2010/63/EU, and after approval of the Local Ethical Committee for animal experiments in Bydgoszcz (27/2016). Animals were housed in individually ventilated cages with environment control (55 ± 10% humidity, 23 ± 2 °C), with a light/dark 12 h/12 h cycle. At the end of the experiment, mice were anesthetized with ketamine (100 mg/kg) and xylazine (10 mg/kg) intraperitoneally. Blood and serum specimens were collected and promptly frozen in liquid nitrogen. Subsequently, upon chest opening, the thoracic and abdominal aortas were extracted, placed in physiological saline, and dissected from the surrounding tissues. For a detailed analysis of transgenic mice response to inflammatory stimuli, in additional experiments, 3-month-old Control and Transgenic mice (n = 5) were used. Animals were then divided into four experimental groups: control and transgenic and LPS-treated control and transgenic mice. Animals were intraperitoneally injected with 10 mg/kg LPS solution (E. coli 055:b5) and the control groups were intraperitoneally injected with an amount of saline equal to the volume of LPS solution used in the experimental groups. At 24h after the LPS injection, mice were anesthetized and intravenously injected with EB. Blood and serum samples, as well as aortic fragments, were collected.

### 4.4. Evaluation of Extracellular Catabolism of Adenine Nucleotides on the Aortic Surface

The aortas were collected, washed with 0.9% NaCl solution, and separated from the surrounding tissues. Aortic fragments were cut longitudinally to expose the endothelial surface and analyzed for the extracellular adenine nucleotide catabolism enzymes activities. Aortic segments were placed in the wells of 24-well plates containing 1 mL of Hanks balanced salt solution (HBSS) and preincubated at 37 °C for 15 min. Substrates suitable for each enzyme were sequentially added to the medium: 50 µM adenosine triphosphate (ATP) for ecto-nucleoside triphosphate diphosphohydrolase (ENTPD1), 50 µM adenosine monophosphate (AMP) for ecto-5′-nucleotidase (E5′NT) and 50 µM adenosine for ecto-adenosine deaminase (EADA). At 0, 5, 15, and 30 min of incubation (37 °C), 50 µL samples were collected. After each substrate incubation, the medium was replaced with a fresh one. During the measurements of ATP and AMP hydrolysis rates, 5µM erythro-9-(2-hydroxy-3-nonyl) adenine (EHNA)—was added to the buffer. Prior to analysis, the samples were centrifuged (20,800× *g*/10 min/4 °C). The conversion of the substrates into the products was quantified using HPLC as previously described [45]. Data are shown in nmol/min/g of tissue.

### 4.5. Determination of Blood Nucleotides and Metabolites Concentration

To determine the concentrations of nucleotides and metabolites, the whole blood samples were instantly frozen using liquid nitrogen Subsequently, they were extracted with 1.3 M HClO4 (ratio 1:1) followed by a centrifugation (20,800× *g*/15 min/4 °C). The resulting supernatants were collected and adjusted to pH 6.0–6.5 using 3 M K3PO4 solution. After 15 min of incubation on ice, the samples were centrifuged again (20,800× *g*/15 min/4 °C), and the supernatants were analyzed with high performance liquid chromatography (HPLC) as previously described [45].

### 4.6. Determination of Amino Acids and Derivatives Concentration

To assess the concentration of plasma amino acids and l-arginine analogs, a portion of plasma (50 µL) was mixed with acetonitrile (ratio 1:2.4) and centrifuged (20,800× *g*/10 min/4 °C). The resulting supernatant was collected and freeze dried. The precipitate was dissolved in water at the same volume as the initial plasma volume. The concentrations of amino acids and their derivatives were measured using high performance liquid chromatography–mass spectrometry (LC/MS). The system consisted of a Surveyor MS autosampler, a Surveyor MS quaternary pump, and a degasser connected to a TSQ Vantage triple quadrupole mass detector. Heated electrospray ionization in positive mode was applied. The column used for separation was a Synergi Hydro-RP 100 with dimensions of 50 × 2 mm and a particle size of 2.5 µm. The mobile phase consisted of water with 5 mM nonafluoropentanoic acid (A) and acetonitrile with 0.1% formic acid (B). The mobile phase was delivered at a rate of 0.2 mL/min and the injection volume was 2 μL. The identity of individual amino acids and the 2-chloroadenosine internal standard was confirmed by fragmentation pattern, chromatographic retention time and the similarity of molecular weights [9].

### 4.7. Intracellular Adhesion Molecule-1 (ICAM-1), Vascular Adhesion Molecule-1 (VCAM-1), Interleukin 6 (IL-6) Measurements

The concentrations of soluble ICAM-1, VCAM-1, and IL-6 in the serum were assessed using enzyme-linked immunosorbent assay kits in accordance with the manufacturer’s instructions.

### 4.8. Evaluation of Endothelial Permeability

Endothelial permeability was quantified by the extravasation of Evans blue (EB) [46]. A 0.5% solution of EB in PBS was injected intravenously. The stain was allowed to circulate for 1h minutes. Mice were sacrificed using an approved injectable anesthetic overdose procedure. Blood and aortas were collected. Blood samples were centrifuged (3000× *g*/10 min/4 °C) and serum were immediately frozen in liquid nitrogen. Aorta fragments were weight and placed in formamide and incubated at 55 °C for 48 h. Aortic tissue extracts were centrifuged (3000× *g*/10 min/4 °C). Evans Blue stain was measured spectrophotometrically at a wavelength of 610 nm. The results are presented as (µg of Evans Blue stain)/(mL of serum or g of tissue).

### 4.9. Statistical Analysis

The results are presented as mean ± SEM. The statistical analysis was performed using using InStat software (GraphPad Prism 8.0, San Diego, CA, USA). The unpaired and paired Student *t*-test was used for comparisons between two groups. Two-way analysis of variance with post hoc Tukey’s test was applied for comparisons of more than two groups. Normality was evaluated by the Shapiro–Wilk test, Kolmogorov–Smirnov test, and the D’Agostino and Pearson Omnibus normality tests. A *p*-value < 0.05 was considered a significant difference.

## Figures and Tables

**Figure 1 pharmaceuticals-16-01409-f001:**
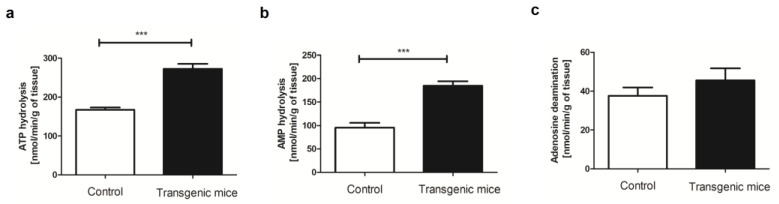
The in vivo usage of tricistronic vector of hHO-1, hE5NT and hENTPD1 results in a significant increase in ATP and AMP hydrolysis on the aortic surface. (**a**) Vascular ATP hydrolysis rates of the control and transgenic mice; (**b**) Vascular AMP hydrolysis rates of the control and transgenic mice and (**c**) vascular adenosine deamination rates of the control and transgenic mice. All values are shown as mean ± SEM (*n* = 10); Student’s *t*-test: *** *p* < 0.001).

**Figure 2 pharmaceuticals-16-01409-f002:**
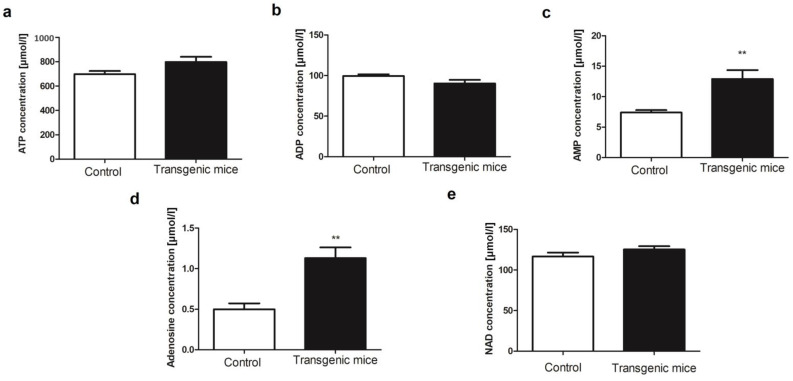
Simultaneous expression of exogenous hHO-1, hE5NT and hENTPD1 results in a two-fold increase in blood adenosine concentration. Blood concentration of (**a**) ATP; (**b**) ADP; (**c**) AMP; (**d**) adenosine; and (**e**) NAD in transgenic and control mice. (All values are shown as mean ± SEM (*n* = 10); Student’s *t*-test: ** *p* < 0.01;).

**Figure 3 pharmaceuticals-16-01409-f003:**
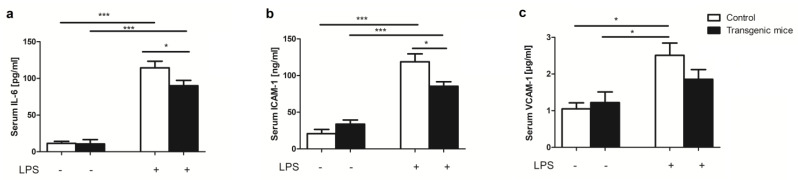
Transgenic mice with simultaneous expression of exogenous hHO−1, hE5NT and hENTPD1 respond better to LPS−induced inflammation. Serum (**a**) IL−6; (**b**) ICAM−1 and (**c**) VCAM−1 concentration in control and transgenic mice after LPS treatment. (All values are shown as mean ± SEM (*n* = 5); Two-way ANOVA with post hoc Tukey test and Student t test: * *p* < 0.05; *** *p* < 0.001).

**Figure 4 pharmaceuticals-16-01409-f004:**
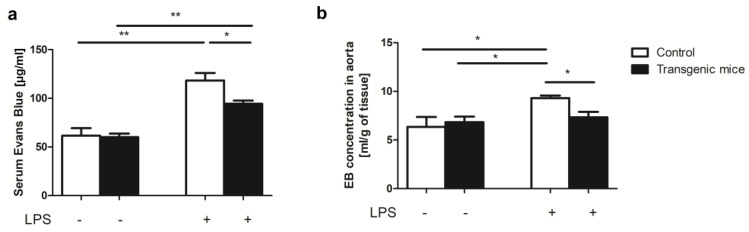
Simultaneous expression of exogenous hHO−1, hE5NT and hENTPD1 has a protective effect on the endothelium after in LPS−induced inflammation. EB concentration in (**a**) serum and (**b**) aortic tissue extract in control and transgenic mice after LPS treatment. (All values are shown as mean ± SEM (*n* = 5); Two-way ANOVA with post hoc Tukey test and Student *t* test: * *p* < 0.05; ** *p* < 0.01).

**Table 1 pharmaceuticals-16-01409-t001:** The in vivo usage of tricistronic construct of hHO-1, hE5NT and hENTPD1 appears to cause slight changes in L-arginine metabolism that may be beneficial to the vascular endothelium. Plasma (a) L-Arginine, (b) Ornithine, (c) Citrulline, (d) Ornithine/L-arginine, and (e) Ornithine/Citruline ratios, as well as (f) asymmetric dimethyl L-arginine (ADMA) and (g) L-NG-monomethyl arginine (L-NMMA) concentration of control and transgenic mice. All values are shown as mean ± SEM (n = 10; Student’s *t*-test: * *p* < 0.05;).

Amino Acid [μmol/L]	Control	Transgenic	*p*-Value
**Arginine**	**84.06 ± 5.18**	**92.65 ± 3.25**	**0.075**
**ADMA**	**0.91 ± 0.05**	**0.78 ± 0.04**	**0.112**
SDMA	0.52 ± 0.09	0.50 ± 0.06	0.868
**L-NMMA**	**0.36 ± 0.01**	**0.28 ± 0.02**	**0.043 ***
Ornithine	140.20 ± 16.48	147.70 ± 17.73	0.777
Citrulline	89.46 ± 2.98	98.48 ± 6.12	0.246
**Ornithine/Arginine**	**1.85 ± 0.05**	**1.32 ± 0.12**	**0.012 ***
Ornithine/Citrulline	1.71 ± 0.13	1.32 ± 0.17	0.131

## Data Availability

Data is contained within the article.

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
