# Peer review of "Endothelial Effects of Simultaneous Expression of Human HO-1, E5NT, and ENTPD1 in a Mouse"

_pharmaceuticals, 2023, doi:10.3390/ph16101409_

Round 1

Reviewer 1 Report

The authors submitted the paper entitled “Endothelial effects of simultaneous expression of human HO-1, E5NT, and ENTPD1 in a mouse” starting from the fundamental argument that vascular endothelium is key target for immune and thrombotic responses that has to be controlled in successful xenotransplantation. The aim was to evaluate in vivo the effects of the simultaneous expression of a three human genes in a mouse generated using the multi-cistronic F2A technology. The obtained results suggested that the human enzymes (hHO-1, hE5NT, hENTPD1), simultaneously encoded in transgenic mice, demonstrated benefits in several biochemical and functional aspects of endothelium, which would be consistent in use of this approach in the context of xenotransplantation.

The paper was very well prepared with exhaustive and scientifically sound introduction, appropriately chosen and executed methodology, as well as high-level-quality discussion adequately comparing so far knowledge of other scientists with the obtained results. Although experimental in nature, this paper presents full clinical relevance, yet to be fully investigated.

Reviewer 2 Report

The authors of this manuscript investigated the effects of the simultaneous expression of three human genes, heme oxygenase 1 (hHO-1), ecto-5'-nucleotidase (hE5NT), and ecto-nucleoside tri-18 phosphate diphosphohydrolase 1 (hENTPD1), in a mouse. The results prove the anti-inflammatory and protective effects of this combination of human genes. It is concluded that genetic modification may be beneficial to preserve endothelial function in the context of xenotransplantation.

The article can be published after some corrections.

1.           The authors use parametric methods for statistical processing of the results. This is acceptable if the data are normally distributed. Was a test for "normality" performed?

2.           Methods section, ELISA reagents. "RAB0477" should be removed; this is a kit for the analysis of TNF, and it was not measured in this work.

3. The authors write

"There was a tendency to higher L-arginine concentration and decrease in its analogues - asymmetric dimethyl L-arginine (ADMA) and L-NG-monomethyl arginine (L-NMMA) in transgenic mice as compared to control". In Table 1 for L-NMMA level p is less than 0.05. Formally, this is not a tendency, but a significant difference.

4.           The number of animals in the groups should be indicated in the captions to Figs. 3 and 4. For uniformity, the colour of the bars should be black and white.

5.           Replace IFN-Y (line 204) with INF-gamma.
